# Biochemical and Genetic Responses of Tea (*Camellia sinensis* (L.) Kuntze) Microplants under Mannitol-Induced Osmotic Stress In Vitro

**DOI:** 10.3390/plants9121795

**Published:** 2020-12-17

**Authors:** Lidiia Samarina, Alexandra Matskiv, Taisiya Simonyan, Natalia Koninskaya, Valentina Malyarovskaya, Maya Gvasaliya, Lyudmila Malyukova, Gregory Tsaturyan, Alfiya Mytdyeva, Marcos Edel Martinez-Montero, Ravish Choudhary, Alexey Ryndin

**Affiliations:** 1Federal Research Centre the Subtropical Scientific Centre of the Russian Academy of Sciences, Sochi 354002, Russia; matskiv.aleksandra@mail.ru (A.M.); taisiya-simony@yandex.ru (T.S.); natakoninskaya@mail.ru (N.K.); lab-bfbr@vniisubtrop.ru (V.M.); m.v.gvasaliya@mail.ru (M.G.); malukovals@mail.ru (L.M.); grisha.tsaturyan@yandex.ru (G.T.); mytdyevaalfia@gmail.com (A.M.); ryndin@vniisubtrop.ru (A.R.); 2Department of Plant Breeding and Plant Conservation, Bioplantas Center, University of Ciego de Avila, Ciego de Avila 65200, Cuba; cubaplantas@gmail.com; 3Division of Seed Science and Technology, ICAR-Indian Agricultural Research Institute, New Delhi 110012, India; ravianu1110@gmail.com

**Keywords:** *Camellia sinensis*, in vitro, osmotic stress, gene expression, micro-plants, culture media, tissue culture

## Abstract

Osmotic stress is a major factor reducing the growth and yield of many horticultural crops worldwide. To reveal reliable markers of tolerant genotypes, we need a comprehensive understanding of the responsive mechanisms in crops. In vitro stress induction can be an efficient tool to study the mechanisms of responses in plants to help gain a better understanding of the physiological and genetic responses of plant tissues against each stress factor. In the present study, the osmotic stress was induced by addition of mannitol into the culture media to reveal biochemical and genetic responses of tea microplants. The contents of proline, threonine, epigallocatechin, and epigallocatechin gallate were increased in leaves during mannitol treatment. The expression level of several genes, namely *DHN2, LOX1, LOX6, BAM, SUS1, TPS11, RS1, RS2,* and *SnRK1.3*, was elevated by 2–10 times under mannitol-induced osmotic stress, while the expression of many other stress-related genes was not changed significantly. Surprisingly, down-regulation of the following genes, *viz*. *bHLH12, bHLH7, bHLH21, bHLH43, CBF1, WRKY2, SWEET1, SWEET2, SWEET3, INV5,* and *LOX7,* was observed. During this study, two major groups of highly correlated genes were observed. The first group included seven genes, namely *CBF1, DHN3, HXK2,*
*SnRK1.1, SPS, SWEET3,* and *SWEET1*. The second group comprised eight genes, *viz*. *DHN2, SnRK1.3, HXK3, RS1, RS2,*
*LOX6, SUS4,* and *BAM5*. A high level of correlation indicates the high strength connection of the genes which can be co-expressed or can be linked to the joint regulons. The present study demonstrates that tea plants develop several adaptations to cope under osmotic stress in vitro; however, some important stress-related genes were silent or downregulated in microplants.

## 1. Introduction

Drought, salinity, and cold are three major factors reducing the growth and yield of many horticultural crops worldwide [1,2]. The negative effects of these factors are explained first of all by the decreasing water potential of plant cells causing the osmotic stress. The osmotic stress leads to oxidative damage and involves the formation of reactive oxygen species (ROS) in plant cells with subsequent membrane damage [1,2]. A series of morphological, physiological, biochemical, and molecular changes that alter the growth, development, and productivity of plants are triggered by osmotic stress. The molecular responses of plants to osmotic stress include perception, signal transduction, gene expression, and ultimately metabolic changes that lead to the stress tolerance [3]. Plants have evolved mechanisms to reduce their oxidative damage by the activation of antioxidant enzymes and the accumulation of osmolytes that effectively scavenge ROS. Numerous genes are involved in a plant’s response to osmotic stress and it is often difficult to analyze the responses in the field or in greenhouse conditions, due to the complex and variable nature of these stresses [2,4,5,6]. This is especially relevant for perennial tree crops, in which response to stress is more complicated compared with grasses.

In vitro stress induction is an alternative tool that can help in the deep investigation of physiological and genetic responses of plant tissues against each stress factor under a controlled environment [2].

Application of osmotically active compounds in tissue culture is a useful method for studying the effects of osmotic stress. Polyethylene-glycol (PEG), mannitol, sorbitol, and NaCl are widely applied to stimulate osmotic stress and study plant responses [4]. As indicated previously, NaCl, mannitol, and sorbitol are able to penetrate into cells and move towards the shoot and leaves. On the other hand, PEG (higher than 3000 MW) cannot pass through the apoplastic barrier of cells—therefore, its osmotic stress is less pronounced, especially in the leaves [4]. Mannitol is a polyhydric alcohol, and the addition of 50–200 mM of mannitol into the culture media decreased growth and increased the sub-culturing period in many crops [5,6,7]. A higher concentration of mannitol reduced the water potential of the nutrient medium and made the transition of water into the plant tissues more difficult causing the osmotic stress [8]. The osmotic potential decreased gradually in plants treated with mannitol, showing that osmolyte accumulation increased over time. Mannitol is commonly used in many investigations to study the osmotic stress response of different plant species in vitro [9,10,11,12,13,14,15]. However, limited information is available on responses under mannitol-induced stress in tree crops, including the tea plant.

The tea plant (*Camellia sinensis* L.) is one of the most important economic tree crops in China, India, Sri Lanka, Kenya, and certain Caucasian countries (Turkey, Georgia, Russia, and Azerbaijan). This perennial, woody, and evergreen crop is grown in more than 60 countries on five continents, from 49° N in Ukraine to 33° S in South Africa [16]. Caucasus tea germplasm collection site (44°36′40″ N, 40°06′40″ E) is located in the border region of the possible tea production and can be the source of the most tolerant genotypes [17]. In most countries, tea plantations are affected by drought conditions that significantly reduce the yield and decrease the distribution of the crop. Due to out-breeding and its long gestation period, the tea plant requires next-generation breeding strategies to improve its drought tolerance. This is possible through a deeper understanding of key regulators and their variants for precision introgressions to achieve better yield and quality under stress conditions [18]. Therefore, different approaches are necessary to deeply understand the tolerance mechanisms to certain stress factors.

Many biochemical and physiological and genetic markers of osmotic-stress response were revealed in many crops, including the tea crop. Among them, sugars, proline, and other osmolytes increase the viscosity of cytoplasm and prevent the membrane damage caused by ROS [19,20]. Recently, transcriptomic studies helped to reveal the differentially expressed genes and candidate markers of stress tolerance in tea plants [21]. Many transcription factors (TFs) and metabolite-related genes have been shown to be involved in osmotic stress response of tea plant. The genes involved in the ABA-independent responsive pathway and the *bZIP*-mediated ABA-dependent pathway participate in tolerance to osmotic stress induced by drought and cold [22]. In tea plants, 12 major TF-families are believed to be involved in the osmotic stress response viz., *AP2/EREBP, bHLH, bZIP, HD-ZIP, HSF (HSP), MYB, NAC, WRKY, zinc-finger protein TFs, SCL, ARR,* and *SPL* [23,24,25,26,27,28,29,30,31,32]. Many sugar metabolism genes were also induced in response to osmotic stress in tea [33]. Most of these genes were revealed using field experimental plots and potted plants in climatic chambers, so validation by an in vitro approach can help in gaining a better understanding of their involvement in the stress response. Tissue culture approaches can be a promising tool for the validation and verification of the revealed genetic markers in tea plants.

Thus, the aim of the current research was to evaluate the osmotic effect caused by mannitol and to assess the physiological and biochemical parameters of tea microplants and expression level of the 31 osmotic stress-responsive tea genes. Most of these genes were selected by us because they were earlier supposed as the informative genetic markers of cold or drought tolerance in tea plant [23,24,25,26,27,28,29,30,31,32,33]. These genes have also been shown to be involved in cold responses in Caucasian tea genotypes [34]. The expression profiles of these genes in tea microplants were analyzed and the correlations between biochemical and genetic responses were established under in vitro mannitol treatment.

## 2. Results

### 2.1. Effect of Mannitol on Physiological and Biochemical Parameters of Tea Microplants

Addition of 200 mM of mannitol into the culture media did not lead to necrotic changes in microplants; however, 300 mM of mannitol caused visible damage to plantlets, such as yellowing and partial drying of leaves in some of them (Figure 1).

The significant decrease in leaf moisture from 66.9 to 52.4% was observed in tea leaves under the mannitol effect. Both concentrations (200 and 300 mM) induced the similar reduction in leaf moisture (Figure 2A) and caused significant elevation in the relative electrical conductivity (REC). However, the highest REC (34.3%) was observed in the presence of 300 mM of mannitol as compared to the control (10.1%) (Figure 2B). A significant decrease in chlorophyll content was observed with 200 mM of mannitol (Figure 2C). The chlorophyll *a* content was decreased 1.8-fold under the Mannitol200 treatment. A similar decreasing trend was observed for chlorophyll *b*. However, the changes in chlorophyll *a* were on the verge of statistical reliability in the treatment with Mannitol300, and there was no significant difference with the control in the level of chlorophyll *b*. The carotenoids content in leaves was also not affected significantly by mannitol (Figure 2D).

Among the 11 amino acids studied, eight showed no difference in concentration as compared to the control treatment (Appendix A). However, proline and threonine contents were elevated, and α-alanine content was decreased significantly when mannitol was added into the culture media. In particular, proline content increased 1.7-fold as compared to the control under the mannitol treatment (Figure 3A). The difference between the two mannitol variants was also non-significant. In addition, threonine content significantly increased three- and four- fold in Mannitol200 and Mannitol300, respectively, compared with the control treatment (Figure 3B). α-alanine content decreased significantly three-fold under the effect of mannitol compared with the control (Figure 3C). Analysis of biochemical indicators of tea quality showed decreasing caffeine content (Figure 3D). However, significant elevation of two catechins, namely epigallocatechin and epigallocatechin gallate, was observed under the effect of mannitol (Figure 3E). The 3.3-fold- and 2.1-fold accumulation of epigallocatechin in leaves was observed in Mannitol200 and Mannitol300, respectively, as compared to the control. The maximum elevation of epigallocatechin gallate was also observed in the Mannitol200, which was 1.5 times higher than the control. Thus, maximum accumulation of epigallocatechin and epigallocatechin gallate in tea leaves was observed when 200 mM of mannitol was added into the culture media. Other catechin contents were not significantly different from the control (Figure 3E). Summarizing these results, it can be concluded that selected concentrations of mannitol in culture media resulted in a decrease in the leaf moisture in tea microplants, and lead to a significant change in cell membrane permeability. In addition, proline, threonine, epigallocatechin, and epigallocatechin gallate were accumulated under the mannitol effect (Figure 3A,B,E). On the other hand, caffeine, chlorophylls and alpha-alanine contents were decreased in the presence of mannitol in culture media (Figure 2C and Figure 3C,D).

### 2.2. Effect of Mannitol on Expression Profile of Osmotic Stress-Related Genes in Tea Microplants

The thirty-one osmotic stress-related genes were separated into three groups (Cluster 1, Cluster 2 and Cluster 3) based on the expression profiles (Figure 4). Cluster 1 included 10 genes (*CBF1, bHLH12, bHLH21, WRKY2, HXK2, SWEET1,2,3, INV5, LOX7*) which, unexpectedly, were downregulated in tea microplants under the mannitol effect. The second cluster combined 12 genes with no changes in expression level in mannitol-supplemented media. Most of the transcription factors (TFs) were either downregulated or not responsive in the plants grown on mannitol-supplemented media. Finally, Cluster 3 included nine genes (*DHN2, BAM, SUS4, RS1, RS2, SnRK1.3, TPS11, LOX1, LOX6*) which showed elevated expression by1.7–12.7-fold in tea leaves under the effect of mannitol. The significant difference between the two mannitol variants was also detected in *LOX1, LOX6, DHN2,* and *RS2* genes. Moreover, the highest expression of *LOX6, DHN2* and *RS2* genes was observed at the greater mannitol concentration.

The correlation analysis regarding mannitol responses showed several significant positive correlations between biochemical compounds and genes in tea microplants (Figure 5). For example, threonine and gallocatechin correlated positively with several genes, such as *RS1, RS2, DHN2, LOX6, BAM5, SnRK1.3, SUS4,* and *HXK3*. In addition, epigallocatechin, epicatechin, and proline correlated positively with two genes, namely *LOX1* and *TPS11*. In addition, alpha-alanine, chlorophylls and carotenoids also correlated positively with the following genes: *CBF1, HXK2, SWEET3, SnRK1.1, SPS,* and *DHN3.* Additionally, epigallocatechin gallate, epicatechin gallate, and caffeine showed significant positive correlations with the *DHN1* and *NAC17* genes.

The positive correlations were also observed between the several genes. For example, *HXK2, SnRK1.1, SPS, DHN3, SWEET3, SWEET1* and *CBF1* correlated positively in response to mannitol. Finally, positive correlations were observed between *RS1, RS2, DHN2, LOX6, SnRK1.3, SUS4, BAM5*, and *HXK3*.

On the other hand, many significant negative correlations were also observed in response to mannitol. For example, threonine and gallocatechin negatively correlated with the following genes: *bHLH12, bHLH21, WRKY2, SWEET*1,2,3, *INV5, LOX7, bHLH43, SnRK1.2, NAC30*, and *HXK1*. In addition, epigallocatechin, epicatechin, and proline negatively correlated with the *NAC17, DHN3, SPS, SWEET3, SWEET1*, and *CBF1* genes. In addition, alpha-alanine, chlorophylls, and carotenoids negatively correlated with the *TPS11*, *LOX1* and *NAC26* genes.

## 3. Discussion

Plants have evolved many responsive mechanisms for the tolerance to water-deficit, such as the maintenance of water-use efficiency, osmotic adjustment, and the protection of the cellular machinery [35]. The goal of our study was to evaluate the effect of high mannitol concentrations on the osmotic stress response of tea microplants in vitro. We select 200 and 300 mM of mannitol to induce medium and severe osmotic stress, respectively, based on our previous experience with tree crops.

Application of osmolytes causes genotype-specific severity of osmotic stress, which is why it should be evaluated for each plant species separately. Our results show that mannitol in culture media induces significant osmotic stress in tea plants in vitro: we observed decreased leaf moisture content by 15%, increased relative electrical conductivity by 20%, and elevated proline content in leaves. These metabolic changes can be seen as evidence of the osmotic stress in tea microplants caused by mannitol in culture media [36].

Chlorophyll and carotenoids contents have been proved to be important indicators of the severity of osmotic stress affecting plants [4]. Chlorophyll *a* content was decreased in the presence of mannitol in culture media. Surprisingly, the highest mannitol concentration showed no significant difference in Chl*b* content compared with the control. No significant deviation was observed in the Chl*a*/Chl*b* ratio. Our results are consistent with some other results in jackfruit and sugar apple plants under drought stress [37]. Nevertheless, most of the published studies showed decreasing pigment contents in plants under the effect of osmotic stress [38]. In the previous studies, this was attributed to the inhibition of chlorophyll synthesis, together with the activation of its degradation by chlorophyllase [39]. In addition, the reduced alteration of the total Chl content can be a reliable marker of tolerant genotypes under water deficit [35]. On the other hand, assessment of the chlorophyll and biochemical compounds in fresh leaves has disadvantages, not only because it may be diluted by a higher plant biomass of the leaves, but also because of the differences in water amount (and hence the dilution effect) within tissue, especially when comparing fully hydrated and partially dried leaf tissues, as in this case. A stress-induced decrease in biomass may, therefore, mitigate the parallel decline in chlorophyll and biochemical compound content. Stress responses may often lead to biomass differences and dilution of certain metabolites or nutrients in non-stressed individuals which can disturb the assessment of their parallel stress-related decrease in treated plants [40].

Among the studied metabolites, proline, threonine, epigallocatechin, and epigallocatechin gallate were accumulated in leaves under the effect of mannitol. In the case of proline, not much research has been focused on understanding the function of the other amino acids in the osmotic stress response. In this study, we evaluated 11 amino acids, but no significant elevation was observed in nine of them (arginine, tyrosine, beta-phenylalanine, leucine, methionine, valine, serine, alpha-alanine, glycine) (data are not illustrated in the article). Many studies reported that free amino acids were accumulated under drought and cold, and mainly serve as mechanisms for osmotic adjustment in plants [37,41,42].

Proline is an indispensable compound in studies related to osmotic stress, and we observed its accumulation in the mannitol-treated tea plantlets, which is in accordance with the other published research [43]. Plant cells have the potential to accumulate proline speedily and degrade it quickly when needed [43]. However, highlighting the role of proline in plant adaptation is not always reliable because in many studies there was no elevation of proline observed in plants under osmotic stress [4]. In addition, some studies reported that proline accumulation was more pronounced under the salt-induced comparing with mannitol- or PEG-induced osmotic stress: the highest proline accumulation was detected in the leaves of NaCl-treated plants, followed by mannitol and sorbitol, and finally by PEG [4].

Along with the other free amino acids, threonine serves as an osmolyte in response to abiotic stress, and is thought to play an important role in plant stress tolerance [44]. Our results are consistent with the other researchers who also showed threonine accumulation in response to osmotic stress in rice [45], wheat [46], and Arabidopsis [42], and also this is consistent with our recent study on tea response to cold [34]. In addition, the recent study showed that threonine accumulated under drought stress only in tolerant genotypes of sesame, so it can be proposed as one of the markers of drought tolerance [47]. Threonine serves as a substrate for isoleucine synthesis. Isoleucine is a branched-chain amino acid which is induced many-fold, along with the other sulfur-containing amino acids, and can play an important role during osmotic stress [45] Surprisingly, we observed the decrease in α-alanine in tea leaves under the mannitol effect. Alanine synthesis is possibly tissue-specific, as was recently shown in Arabidopsis: alanine was elevated in flowers under drought stress but not in leaves [45]. The alanine fermentative pathway does not regenerate NAD+, and unlike several metabolites known for either their signaling or protective role upon abiotic stresses—e.g., proline, betaine [48]—alanine intrinsically may not play a protective role like other osmolytes. However, in some abiotic stresses, alanine plays the role of a carbon storage pool [48]. Most stress-related organic compounds are secondary plant metabolites, and tea (*Camellia sinensis*) contains large amounts of polyphenols, mainly catechins, that belong to the flavan-3-ol class. Catechins are the major components, occupying more than 10% of dry tea leaves and responsible for the tea quality [49]. Epigallocatechin and epigallocatechin gallate were found to be the highest fractions among the other catechins in tea plants [50], which corresponds with our results on in vitro tea plants. Moreover, we observed significant accumulation of epigallocatechin and epigallocatechin gallate under the mannitol treatment. Our results are consistent with Hernández et al. (2004), who showed that epigallocatechin gallate and epicatechin gallate concentrations increased progressively during drought in *Cistus clusii*, reaching maximum values after 30 days of stress [51]. The other studies also showed that epigallocatechin gallate promoted tolerance to osmotic stress by the mitigation of oxidative stress through efficient ROS scavenging by the enhanced activity of antioxidant enzymes. In addition, a recent study on tea plants showed a decrease in most of the biochemical substances responsible for tea quality; however, the authors also revealed that the content of EGCG and ECG in *C. sinensis* leaves tended to first decrease after 2 days of drought and then increase after 5 days of drought stress [11]. Other researchers also reported that phenolic biosynthesis in tea is significantly enhanced by osmotic stress [52]. Moreover, wide fluctuations in the content of catechins between the different tea clones were reported in response to drought [53,54]. Studies in the other plant species showed that some polyphenolic compounds are accumulated under the water deficiency in plants and play a protective role against osmotic stress [41,55,56]. Moreover, the role of exogenous EGCG in stimulating plant tolerance to several abiotic stresses due to the high antioxidant potential and/or flavonoid signaling-mediated stress response was recently demonstrated in several plant species [57,58,59].

The set of the important transcription factors and metabolite genes that were confirmed to be important in tea drought and cold and the other abiotic stress-responses [23,24,25,26,27,28,29,30,31,32,33] was selected for this study to validate them under mannitol-induced osmotic stress. The nine genes (*DHN2, BAM, SUS4, RS1, RS2, SnRK1.3, TPS11, LOX1, LOX6*) with significantly elevated expression of 1.7–12.7-fold in tea leaves were revealed under the mannitol treatment. The highest expression levels of *LOX6, RS2,* and *DHN2* genes were observed at the greater mannitol concentration, confirming the direct relationships between the strength of osmotic stress and the level of expression of these three osmotic stress-related markers. Significantly increased expression of *DHN2* gene under mannitol treatment confirms the water deficiency in tea microplants, which is the trigger for *DHNs*. *DHN1,2,3* are transcription factors from the LEA superfamily, encoding the cryoprotective proteins acting as chaperons, playing a critical role in the protection of plants during abiotic stresses [17,60]. Earlier, in our studies [34] and other studies [27], it was shown that *DHN2* can serve as the marker for cold-tolerant cultivars of tea plants. Our result on *DHN2* is consistent with the earlier studies and can be another piece of evidence of the osmotic stress in tea microplants caused by mannitol.

Three genes (*CsLOX1, CsLOX6,* and *CsLOX7*) of the lipoxygenase gene family were earlier suggested to be involved in ABA-independent responses to abiotic stress in tea plants [61]. These genes are involved in lipid catabolism and jasmonate and oxlipin synthesis [62]. We observed the upregulation of two genes, *LOX1* and *LOX6*, in response to mannitol, indicating the activation of the lipid peroxidation processes in tea leaves under mannitol treatment.

Six upregulated genes (*BAM, SUS4, RS1, RS2, SnRK1.3* and *TPS11)* are related to the sugar metabolism. Thus, we can suggest that mannitol treatment activates the sugar metabolism in tea plants under osmotic pressure. *BAM* gene is a key regulator which hydrolyzes starch to maltose and plays an important role in the cold stress response in tea and in photosynthetic protection under osmotic stress [33]. *SUS* genes encoded sucrose synthase (Sus), a key enzyme of sucrose metabolism. Previous studies reported that the transcription levels of *Sus1* increased after exposure to cold and drought stress conditions [33,63]. *CsSUS* genes have a bidirectional role in the conversion of sucrose to glucose and fructose, which is necessary for sucrose balance in the plant [20]. *RS* genes are the key regulators in raffinose synthesis, and the over-expression of these enzymes encoding genes led to an increase in raffinose and was shown to be upregulated under cold stress in tea plant [20]. Additionally, *TPS* is the key controller gene of trehalose synthesis, which was shown to play a critical role in plant development and the stress response [64]. In tea plants, *CsTPS11* was significantly induced by cold stress [33]. In our study, elevated expression of this gene was observed, suggesting its important role in the drought response of tea plants. *CsSnRK1* is a serine/threonine protein kinase, and these genes act as key regulators involved in sugar signaling. Earlier, it was suggested that alterations in sugar content stimulate signaling to mediate stress-related gene transcription and integrate ABA in response to stress [33]. In our study, only *SnRK1.3* was upregulated in response to mannitol treatment, suggesting the specific role of this gene in the drought response of tea plants.

Surprisingly, ten genes *(CBF1, bHLH12, bHLH21, WRKY2, HXK2, SWEET1,2,3, INV5,* and *LOX7)* in our study were downregulated in tea under the mannitol treatment. Among them, we can see the regulator (*CBF1*) of ABA-independent stress response, three transcription factors (*bHLH12, bHLH21* and *WRKY2)* for ABA-dependent stress response, four genes (*HXK2, SWEET1,2* and *3*) participating in sugar signal transduction, and two other genes (*LOX7* and *INV5*) for lipid and sugar metabolism. All these genes were proposed in recent transcriptomic studies as the possible drought-responsive markers in tea plants. For example, *CsWRKY2* was shown to be the marker of cold- and drought-response, activating ABA-signaling [26]. In addition, *bHLH12* and *bHLH21* were proposed as drought stress-responsive genes in tea, activating ABA-signal transduction [30]. *HXK* is a sensor of glucose in plants, but it is also involved in glucose metabolism, and *CsHXK2* was significantly suppressed with glucose accumulation [33]. In addition, three sugar transporters (*SWEET1, SWEET2* and *SWEET3*) were downregulated, indicating the inhibition of the sugar flux in the intercellular space during in vitro osmotic stress.

Many genes which were included in our study were suppressed or silent. The one of the possible explanations for these results may be the duration of the stress exposure. In our experiments, analyses were performed after long-term exposure to mannitol for up to 2 months. It can be assumed that some osmotic stress-related genes are more actively transcribed at the initial stage of stress. Other genes are expressed at high levels under long-term stress exposure. On the other hand, in many cases, DNA-methylation was the reason for the inhibited expression of different genes in plants cultured in vitro [65,66]. Abiotic stresses like those induced by environmental conditions seem to have an effect on DNA methylation and organogenesis. It has been proven that DNA-methylation can be inherited by subsequent generations. The stability of DNA methylation patterns through in vitro culture and ex vitro acclimatization has been observed in several species [67]. One possible mechanism involving the alteration of DNA methylation patterns is the activation of transposable elements. Evaluation of correlations between the genes and downstream biochemical processes can help to identify gene modules whose expression profiles are very similar. This can help in gaining a better understanding of the mechanism of the response and adaptation of crops to each stress factor. The modules of densely inter-connected genes can be analyzed by searching for patterns in connection strength [68]. A high level of correlation indicates high strength connection of the genes which can be co-expressed or can be linked to the joint regulons [69]. This information can be provided through the analysis of the expression profiles of each gene through the different treatments. In our study, the correlations were analyzed to identify gene expression patterns and their regulation of downstream physiological indices of drought in tea plants. Co-expression of several genes observed under the mannitol effect demonstrates their similar pattern under osmotic stress in tea plants. We found two groups of highly correlated genes. The first group included *CBF1, DHN3, HXK2, SnRK1.1, SPS, SWEET3,* and *SWEET1*, and second group comprised the following genes, namely*DHN2, SnRK1.3, HXK3, RS1, RS2, LOX6, SUS4,* and *BAM5*. These results indicate that the mentioned genes have a similar expression character during drought stress. Co-expression of these genes suggesting shared upstream pathways for signal transduction and regulation under osmotic pressure. Highly correlated gene modules with specific expression patterns can help in illustrating the framework of the drought stress transcriptome.

Further studies with a greater range of treatments (e.g., timing, severity, frequency) and different osmotic agents (PEG, Sorbitol, etc.) need to be examined in future studies to provide more clues for understanding the adaptation and tolerance mechanisms in tea plants.

## 4. Materials and Methods

### 4.1. Plant Material and Osmotic Stress Induction

The plant material for this study was obtained from the germplasm collection of the Subtropical Scientific Centre of the Russian Academy of Sciences, Sochi, Russia. The microplants of the best local genotype, *Kolkhida*, propagated vegetatively in vitro for 2 years, were used in this study. Microplants were propagated on the ½ MS [70] culture media supplemented with benzylaminopurine (6 mg/L), naphthyl acetic acid (1 mg/L), gibberellic acid (2 mg/L), sucrose (25 g/L) and solidified with agar (8 g/L) after adjusting pH to 5.7 [71]. Microplants 2.0–3.0 cm in height with the 3–5 fully developed leaves were used as explants for the experiments.

The culture media composed by ½ MS + NAA 0.5 mg/L, agar 8 g/L, sucrose 25 g/L, pH 5.7 were used for the control treatment. The experimental media were the same but were supplemented with 200 or 300 mM of mannitol. Microplants were cultured in the glass vessels of 150 mL, with 20 mL of the culture media in each, where 3 plants per vessel were cultured for 2 months before the sampling. Cultures were grown in the climatic room under the temperature of 22–25 °C (with an illumination regime of 16 h of light and 8 h of dark, with light intensity of 54 µmol m^−2^ s^−1^). Experimental treatments with these plants were repeated twice in the years of 2018 to 2019; in total, 300 vessels with 3 plants in each were included for each year of the study. The leaves of 9 tea plantlets were collected randomly from each treatment and used for the RNA extraction and physiological analyses.

### 4.2. Physiological Analyses

Relative electrical conductivity (REC) was measured using a portable conductivity meter ST300C (Ohaus) to assess the electrolyte leakage, indicating the damage of leaf tissues. The leaf samples (the whole leaves from the 5–6 plantlets) of 300 mg in total were immersed in 150 mL of deionized water at a temperature 20–22 °C. The measurement of electrical conductivity was done immediately after the leaves’ immersion (L1) and two hours later (L2). The relative electrical conductivity (REC %) was calculated as: $REC\;\left(\%\right)=\frac{L1}{L2}*100$ [72]. Five biological and three technical repetitions were used for this analysis (n = 90).

The leaf moisture content (LM) was evaluated to assess the water decrease in leaf tissues under the effect of mannitol: about 1 g of mixed probe of the fresh leaves was weighed (FW) and then dried at 105 °C for five hours and weighed again (DW). Leaf moisture content was calculated according to the formula: $LM\left(\%\right)=\left(\frac{\left(FW-DW\right)}{FW}\right)*100$ [73]. Three biological and three technical repetitions were used for this analysis (n = 60).

Chlorophyll and carotenoid contents (mg/g fresh leaf mass) were evaluated spectrophotometrically. For this, 170 mg of leaf sample (mixed probe from several plantlets) was homogenized and transferred into a conical flask and extracted with 25 mL of ethanol (95%) with the assistance of a magnetic stirrer (magnet size 4.0 × 0.5 cm) at 700 rpm for 15 min at room temperature (20 ± 2 °C). The supernatant was separated by decanting. Residues were extracted again using the same procedure. Extracted supernatants were combined and filtered through the filter paper. Determination of chlorophyll a (Chl a), chlorophyll b (Chl b), and total carotenoids (Ca) was performed using a spectrophotometer PA-5400vi (Russia) at various wavelengths (665, 649 and 440.5 nm for Chl *a*, Chl *b, and* Ca, respectively). All determinations were performed in five biological and three technical replicates (n = 60). The concentrations of chlorophyll *a*, *b* and total carotenoids (mg/mL) were calculated by the Smit–Benitez formula as published by Shlyk et al. [74]. The equations used for the quantification are (i) *C_Ch__a_* = 13.36*A*_665_ − 5.19*A*_649_; (ii) *C_Ch__b_* = 27.43*A*_649_ − 8.12*A*_665_; (iii) *C_Ca_* = (1000*A*_440.5_ − 2.13*C_Ch__a_* − 97.63*C_Ch__b_* )/209. The results were recalculated and expressed as mg/g of fresh leaf mass.

Amino acid contents (arginine, tyrosine, beta-phenylalanine, leucine, methionine, valine, threonine, proline, serine, alpha-alanine, and glycine) (µg/g fresh leaf mass) were evaluated spectrophotometrically using Kapel-105M (Russia). For this analysis, fresh tea leaves (300 mg) were ground and extracted with 10 mL 75% (*v*/*v*) alcohol for 10 min, and the mixture was centrifuged at 12,000 rpm and 4 °C for 15 min. The supernatant was collected for the pre-column derivatization procedure: tea extract (180 μL) was mixed with 100 μL dinitrofluorobenzene solution (10 mg ml−1, *w*/*v*), 100 μL NaHCO3 (0.5 M, pH 9.0) and 20 μL ddH2O in a 1.5-mL centrifuge tube. The mixture was placed in a water bath at 60 °C for 60 min in the dark. After returning to room temperature, 400 μL KH2PO4 (0.01 M, pH 7.0) was added to the tube, vortexed, and kept in the dark for 15 min. Reference solutions of each of the amino acids were prepared by the same method. The mixtures were filtered through a 0.22 μm Millipak filter before capillary electrophoresis analysis [75]. This analysis was conducted in three biological and three technical replicates (n = 60).

Caffeine and catechin (mg/g fresh leaf mass) contents were evaluated by HPLC using the ethanol extraction protocol: 300 mg of fresh leaves were grinded in 10 mL of ethanol 80% and incubated in a water bath for 15 min at 60 °C, sonicated for 5 min, and then cooled. The residue was extracted twice more with the same solvent. The combined supernatants were then filtered through a 0.45 μm Millipore nylon filter. HPLC was carried out on a Milichrom liquid chromatographer (EcoNova, Novosibirsk, Russia) with an autosampler. The gradient system consisted of a mixture of acetonitrile and 20 mM KH2PO4. The flow rate was 1 mL/min. The tea extract (10 μL) was injected into the column. The initial composition of the mobile phase, consisting of 7% (*v*/*v*) solvent A (100% acetonitrile) and 93% solvent B (20 mM KH2PO4), was maintained for 7 min. Solvent A was then increased linearly to 10% at 20 min, 15% at 25 min, 20% at 30 min, and 25% at 45 min to 70 min. Programming was then continued in the isocratic mode as follows: 40% A at 70.1 to 75.0 min and 7% A at 75.1 to 90.1 min. This analysis was conducted in three biological and three technical replicates (n = 90).

### 4.3. Gene Expression Analysis

Total RNA was extracted from 200 mg fresh leaves using the CTAB protocol [76] with minor modifications: mercapthoethanol and proteinase K treatment steps were removed. The concentration and quality of RNA were determined using BioDrop µLite spectrophotometer and RNA-integrity was assessed by agarose gel electrophoresis. RNA samples were dissolved and treated with DNase I using a commercial manual (Biolabmix, Russia, http://biolabmix.ru/), and reverse transcription was performed using the MMLV-RT kit according to the manual instructions (Biolabmix, Novisibirsk, Russia, http://biolabmix.ru/). The efficiency of DNaseI treatment and reverse transcription was tested by agarose gel electrophoresis and by qRT-PCR. Only those samples that confirmed the absence of genomic DNA contamination were included in further analysis of gene expression. In total, 31 osmotic stress-related genes of *Camellia sinensis* were included in the analysis. The genes and the primer sequences are listed in Appendix A. Actin [77] was taken as a reference gene and results were quantified using a Light Cycler 96 analyzer (Roche, Japan). qPCR was performed in Roche LightCycler 96-well plates. Each reaction mixture contained 7.5 μL of 2 × SybrGreen buffer BioMaster HS-qPCR (Biolabmix, Novisibirsk, Russia, http://biolabmix.ru/), 1 μL diluted cDNA, 0.2 μL of forward primer (10 μM), 0.2 μL of reverse primer (10 μM), and 6.1 μL of ddH_2_O in a total volume of 15 μL. The following amplification conditions were applied for all genes: 1 cycle at 95 °C for 5 min, 40 cycles at 95 °C for 10 s, 60 °C for 30 s, followed by 1 cycle of 72 °C for 120 s, 60 °C for 180 s and 97 °C for 1 s. RNase-free water was used as a negative control. In our analyses, each biological sample had three biological replicates, and each biological replicate had three technical replicates. The relative gene expression level was calculated by the Livak and Schmittgen [78] using the following algorithm: 2^-ΔΔCq^, where:ΔΔCq = (Cq_*gene of interest*_ − q_*internal control*_) _*treatment*_ − (Cq_*gene of interest*_ − Cq_*internal control*_) _*control*_

### 4.4. Statistical Analysis

All analyses were repeated twice with three to five biological replications and three technical replicates. Statistical analyses were carried out using XLSTAT software (Addinsoft, New York, NY, USA). Student’s t-test, Spearman’s correlation tests, and Ward’s hierarchical clustering were performed to evaluate data and confirm the significant differences (at the level *p* < 0.05) between the genes expression profiles and biochemical substances.

## 5. Conclusions

To conclude, the key findings of the current research are:(i)Threonine, proline, epigallocatechin, and epigallocatechin gallate were accumulated in response to mannitol and can be important markers for tea drought tolerance.(ii)Out of 31 studied genes, nine genes (*DHN2, BAM, SUS4, RS1, RS2, SnRK1.3, TPS11, LOX1* and LOX6) were significantly upregulated in tea leaves under the mannitol effect, and six of them are related to the sugar metabolism. These genes can be proposed as the reliable markers of drought response in tea plant.(iii)Several transcription factors (*CBF1, bHLH12, bHLH21, WRKY2)* and sugar-signaling genes *(HXK2, SWEET1,2,3)*, as well as metabolite genes *(INV5, LOX7*), were down-regulated in tea leaves under the mannitol effect, which can be related either to the stress exposure period or to their suppression under in vitro conditions.(iv)The similar expression character under both mannitol treatments was observed in the two groups of genes: the first group comprised seven genes (*CBF1, DHN3, HXK2, SnRK1.1, SPS, SWEET3,* and *SWEET1*) and second group included eight genes (*DHN2, SnRK1.3, HXK3, RS1, RS2, LOX6, SUS4,* and *BAM5*). These two modules of densely inter-connected genes can possibly have a high level of co-expression under the mannitol effect.

These findings will be useful for evaluation of the reproducibility of stress-tolerance markers in different tea genotypes. Further studies with a greater range of treatments (e.g., timing, severity, frequency) and different osmotic agents (PEG, Sorbitol, etc.) are necessary to provide more clues for understanding the adaptation and tolerance mechanisms in tea plants to osmotic stress.

## Figures and Tables

**Figure 1 plants-09-01795-f001:**
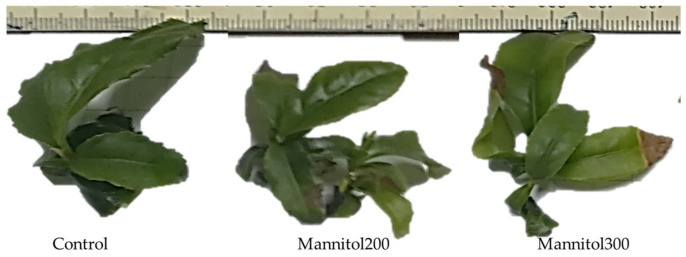
Experimental tea microplants under the effect of mannitol in culture media.

**Figure 2 plants-09-01795-f002:**
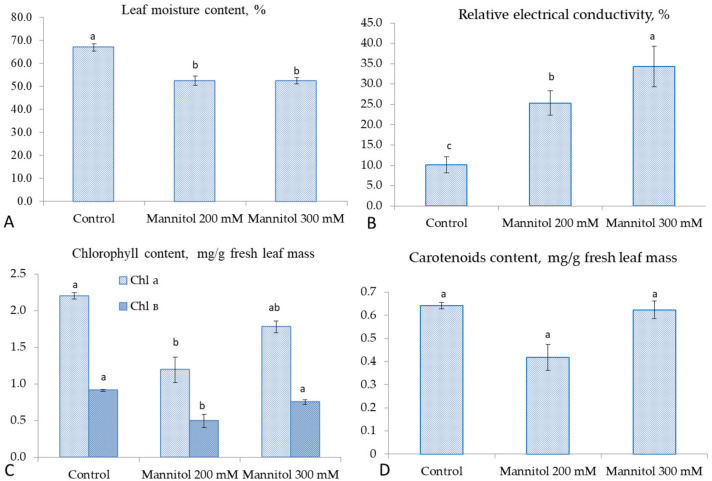
Changes in physiological parameters in tea microplants under the effect of mannitol in culture media: (**A**) leaf moisture content (%); (**B**) relative electrical conductivity of leaf tissues (%); (**C**) chlorophylls content (mg/g fresh leaf mass); (**D**) carotenoids content (mg/g fresh leaf mass). Bars represent standard deviations, and different small letters indicate significant differences among means at *p* < 0.05 (n = 60 − 90).

**Figure 3 plants-09-01795-f003:**
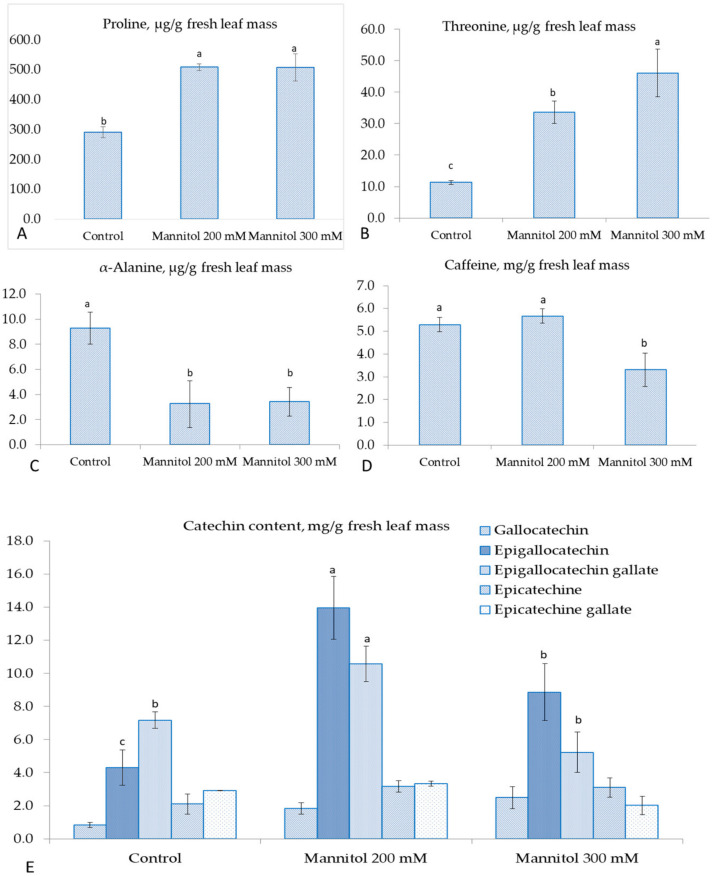
Changes in biochemical parameters in tea microplants under the effect of mannitol in culture media: (**A**) proline content in fresh tea leaves (µg/g); (**B**) threonine content in fresh tea leaves (µg/g); (**C**) α-alanine content in fresh tea leaves (µg/g); (**D**) caffeine content in fresh tea leaves (mg/g); (**E**) catechins content in fresh tea leaves (mg/g). Bars represent standard deviations, different small letters indicate significant differences among means at *p* < 0.05 (n = 60 − 90).

**Figure 4 plants-09-01795-f004:**
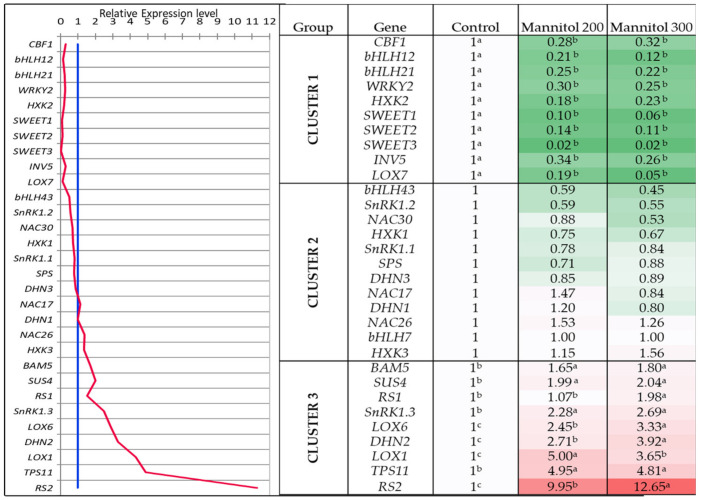
Genes expression profile and the heat map of the relative gene expression levels in tea microplants under the effect of mannitol 200 and 300 mM. Different small letters indicate significant differences at *p* < 0.05 (n = 60).

**Figure 5 plants-09-01795-f005:**
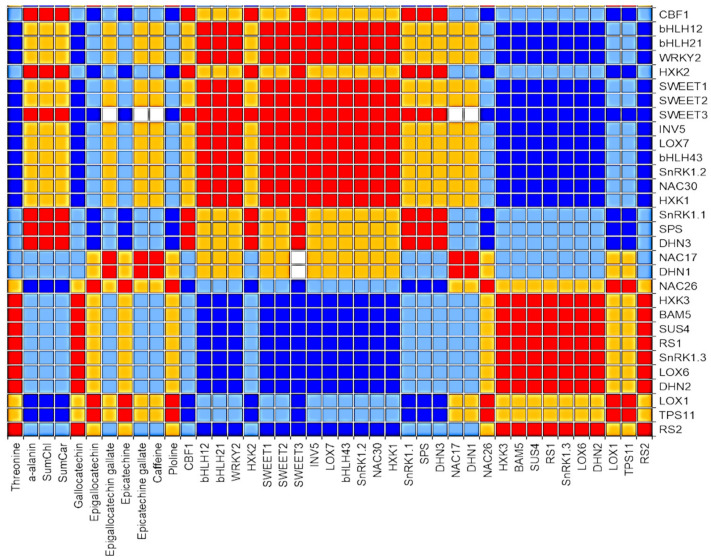
Correlation heat map of biochemical parameters and stress-related genes:red cells—significant positive correlations; dark-blue cells—significant negative correlations. Yellow and light blue cells—non significant positive and negative correlations, respectively.

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
