# Peer review of "Biochemical and Genetic Responses of Tea (*Camellia sinensis* (L.) Kuntze) Microplants under Mannitol-Induced Osmotic Stress In Vitro"

_plants, 2020, doi:10.3390/plants9121795_

Round 1
Reviewer 1 Report
The manuscript describes a study to evaluate the effect of the osmotic stress induced by addition of mannitol into the culture media on tea microplants. With this aim, some physiological and biochemical parameters and the expression level of several stress-responsive genes in tea microplants were determined to unravel the correlations between physiological and genetic responses under in vitro mannitol treatment. The manuscript contains some interesting information but it suffers from some important flaws as noted below.
I do not understand how the authors have determined the relative water content (RWC) value. Indeed, the formula reported in lines 305-307 is used to determine the percentage of water in leaf samples and not the RWC. This parameter represents the water content (as percentage) at a given time as related to the water content at full turgor. Hence, to determine RWC is necessary to evaluate the turgid weight (TW) after hydration of the leaf samples, using the following equation: RWC (%) = [(FW-DW) / (TW-DW)] x 100, where FW and DW are the fresh and dry weight, respectively, and TW the turgid weight. Please, clarify this aspect in material and methods and throughout the manuscript.
Lines 95-96: how the authors explain the different effect of mannitol treatment on leaf chlorophylls? Indeed, they say that “300 Mmol of mannitol caused visible damage of plantlets such as yellowing and partial drying of leaves” (lines 86-87 and Figure 1). If the leaves of the Mannitol 300 treatment were, at least partly, dry and yellow, I suppose that they should contain a lower amount of chlorophylls. Moreover, why the authors did not show the individual chlorophylls a and b and, hence, the Chl a/Ch b ratio? This ratio could be useful to discuss the changes in the pigment composition and related photosynthetic response, being Chl b mainly presents in the LHC and the Chl a in the reaction centres of photosystem II.
The authors should specify in both the text and figures if the biochemical results are expressed on g of fresh weight (g FW) or dry weight (g DW). Only in Figure 3, the authors indicate that results are expressed on g of fresh leaf but in all the other parts of the manuscript (especially Material and Methods, Figure 1 and Results) it is not specified. The authors should specify the units throughout the manuscript, because the results could be strongly affected by this aspect (see also below).
In the discussion there are some speculative parts. For example, at lines 177-179 the authors say that “These results are showing that osmotic stresses lead to over production of reactive oxygen species (ROS) causing inhibition of photosynthesis in accordance with the other studies”. The results shown by the authors only suggest (but not demonstrate) this, for this purpose the authors should measure the ROS production and the photosynthetic performance of the plants. Moreover, the authors discuss the chlorophylls content saying that “Also, chlorophyll contents were decreased in the presence of mannitol in culture media”. The results shown by the authors indicate a decrease of chlorophylls at 200 mM, but the absence of significant changes at the highest concentration of mannitol (300 mM). The authors should specify and discuss this result (please, see also the above comments).
In the discussion, the authors compare their results with those of Wang et al. (2016). However, the last authors expressed the values of metabolic compounds, such as aminoacids and caffeine, on a leaf dry weight basis (g g DW-1), while in the present paper (at least from what I suppose looking at Figure 2) the data were referred on a leaf fresh weight basis (g g FW-1). As reported before, the authors should specify the units used in this work throughout the manuscript. Moreover, I suggest the authors to calculate the concentration of different compounds also on leaf dry weight and to verify if (and how much) the different water status of treated and untreated plants affects the results. Indeed, the concentrations of metabolic compounds could be strongly affected by the lower water content of leaf tissues treated with mannitol compared to control.
The graphical quality of the figures 1 and 2 must be improved. The authors should uniform the style used in the figures (i.e. the bar color for different treatment, units, size font, number of decimals after the comma in Figure 3C, etc.). I do not understand the units used for Threonina, Alanine and Proline reported in Figure 3 (mkg/g fresh weight?). Moreover, In the legend of the figures, the authors should clearly indicate what represent the bars (i.e., means ± SE or SD? and the number of samples, n=?) and the meaning of different letters (i.e., significant differences among means and p value).
Finally, the article shows some grammatical errors and sentences to be rephrased, so I suggest a moderate revision of English (some examples: lines 95-96; 107-109; 114-115; 197; 202.204; 268-270, etc.).
Minor points
Lines 81-82: “Our results showed the silencing or downregulation of many stress-responsive genes in tea microplants under the mannitol treatment”. This sentence is more suitable for the results and/or discussion sections. In the introduction the authors should report the aims of the work and not the results obtained, i.e. “The expression level of several stress-responsive genes in tea microplants was determined to unravel the correlations between physiological and genetic responses under in vitro mannitol treatment”.
Lines 122-124: This sentence is speculative because the authors did not measure the water potential.
Line 161: Please, replace “negatively” with “negative”
Line 225: Please, replace “activate” with “activates”
Line 232: Please, separate “raffinose” and “synthesis”
Author Response
Dear Reviewer,
Thank you for the careful examination of our results. We are really appreciating this. We agree with the all suggestions and we are really grateful for your work with our article. We have tried to do your best to follow your suggestions. We increased the reference list by more than 25 new references and compared our results with them. However, if you are not fully satisfied with our responses, we are ready to complete or revise the manuscript more.
Reviewer 1
I do not understand how the authors have determined the relative water content (RWC) value. Indeed, the formula reported in lines 305-307 is used to determine the percentage of water in leaf samples and not the RWC. This parameter represents the water content (as percentage) at a given time as related to the water content at full turgor. Hence, to determine RWC is necessary to evaluate the turgid weight (TW) after hydration of the leaf samples, using the following equation: RWC (%) = [(FW-DW) / (TW-DW)] x 100, where FW and DW are the fresh and dry weight, respectively, and TW the turgid weight. Please, clarify this aspect in material and methods and throughout the manuscript.
- Thank you for paying attention. We agree with this and we have clarified and corrected this point in the article.
Lines 95-96: how the authors explain the different effect of mannitol treatment on leaf chlorophylls?
- This result for us was also unexpectable. We have just can suggest the different reaction to the different stress severity. For example, you can see in the study of Wang et al. 2016 that 2-days drought leaded to the decrease of some catechins, on the other hand, 5-days drought resulted elevation of these catechins comparing with the control treatment. Here we are discussing not catechins, but the chlorophyll. We presented our results as it is. However, if you suggest that the figure should be removed and the additional study on Chl is necessary we can think about it.
Indeed, they say that “300 Mmol of mannitol caused visible damage of plantlets such as yellowing and partial drying of leaves” (lines 86-87 and Figure 1). If the leaves of the Mannitol 300 treatment were, at least partly, dry and yellow, I suppose that they should contain a lower amount of chlorophylls.
- Concerning your note about the photos and leaf color – yes, looking at the picture we can see that the leaf color is different. However, not all the plantlets had the similar colors and leaf damage. We selected for the fig 1 the most typical phenotypes of the experimental plants. However, to evaluate the chlorophyll content we used the mixed leaf samples and statistical evaluation of the differences showed the results as it is.
Moreover, why the authors did not show the individual chlorophylls a and b and, hence, the Chl a/Ch b ratio? This ratio could be useful to discuss the changes in the pigment composition and related photosynthetic response, being Chl b mainly presents in the LHC and the Chl a in the reaction centres of photosystem II.
- According to your advice we changed the diagram of chlorophyll content. Now it is contains separate data on chl a and chl b. We added the description in the Results as well. However we observed no significant changes in the ratio of Chla/Chlb.
The authors should specify in both the text and figures if the biochemical results are expressed on g of fresh weight (g FW) or dry weight (g DW). Only in Figure 3, the authors indicate that results are expressed on g of fresh leaf but in all the other parts of the manuscript (especially Material and Methods, Figure 1 and Results) it is not specified. The authors should specify the units throughout the manuscript, because the results could be strongly affected by this aspect (see also below).
- Thank you for this note. We have followed your suggestion
In the discussion there are some speculative parts. For example, at lines 177-179 the authors say that “These results are showing that osmotic stresses lead to over production of reactive oxygen species (ROS) causing inhibition of photosynthesis in accordance with the other studies”. The results shown by the authors only suggest (but not demonstrate) this, for this purpose the authors should measure the ROS production and the photosynthetic performance of the plants.
- We agree with this comment and removed this sentence
Moreover, the authors discuss the chlorophylls content saying that “Also, chlorophyll contents were decreased in the presence of mannitol in culture media”. The results shown by the authors indicate a decrease of chlorophylls at 200 mM, but the absence of significant changes at the highest concentration of mannitol (300 mM). The authors should specify and discuss this result (please, see also the above comments).
- The reviewer is right. We have tried to clarify this part in the Discussion.
In the discussion, the authors compare their results with those of Wang et al. (2016). However, the last authors expressed the values of metabolic compounds, such as aminoacids and caffeine, on a leaf dry weight basis (g g DW-1), while in the present paper (at least from what I suppose looking at Figure 2) the data were referred on a leaf fresh weight basis (g g FW-1). As reported before, the authors should specify the units used in this work throughout the manuscript.
- We have tried to follow your recommendations. We added more references in the discussion.
Moreover, I suggest the authors to calculate the concentration of different compounds also on leaf dry weight and to verify if (and how much) the different water status of treated and untreated plants affects the results. Indeed, the concentrations of metabolic compounds could be strongly affected by the lower water content of leaf tissues treated with mannitol compared to control.
- In our future studies we will follow your recommendation to avoid the measurement of biochemical substances in wet tissues and to recalculate each parameter to the dry leaf weight. However, some substances are oxidized and catabolized during desiccation of the leaves. On the other hand, many studies used fresh leaf content of evaluated biochemical substances. Here you can see some of them:
- Guo et al. 2020 doi:10.3390/plants9040520 - metabolites
- Darko et al. 2019 https://doi.org/ 10.1371/journal.pone.0226151 - Determination of metabolites from leaf and root sap
- Joshi et al. Amino Acids (2010) 39:933–947 DOI 10.1007/s00726-010-0505-7 – amino acids in fresh leaf weight
- Medeiros et al. 2012 Braz. J. Plant Physiol., 24(3): 181-192, 2012 - free aminoacids in fresh leaves
- Kabbadj et al. 2017 https://doi.org/10.1371/journal. pone.0190284 – chlorophyll and amino acids were evaluated in fresh leaves
- Fuzy et al. 2019 https://doi.org/10.1007/s11738-019-2842-9 - chlorophyll evaluated in fresh leaves
- Bolat et al. 2014 https://doi.org/10.1155/2014/769732 – chlorophyll and amino acids were evaluated in fresh leaves
- Rodrigues et al. 2010 – Revista Ciência Agronômica, 41(2):245-252 - biochemical compounds
The graphical quality of the figures 1 and 2 must be improved. The authors should uniform the style used in the figures (i.e. the bar color for different treatment, units, size font, number of decimals after the comma in Figure 3C, etc.). I do not understand the units used for Threonin, Alanine and Proline reported in Figure 3 (mkg/g fresh weight?).
- We followed all your suggestions and we have tried to improve and unify the figures style.
Moreover, In the legend of the figures, the authors should clearly indicate what represent the bars (i.e., means ± SE or SD? and the number of samples, n=?) and the meaning of different letters (i.e., significant differences among means and p value).
- The figures are supplemented with these details
Finally, the article shows some grammatical errors and sentences to be rephrased, so I suggest a moderate revision of English (some examples: lines 95-96; 107-109; 114-115; 197; 202.204; 268-270, etc.).
- Hope now it is better
Minor points
Lines 81-82: “Our results showed the silencing or downregulation of many stress-responsive genes in tea microplants under the mannitol treatment”. This sentence is more suitable for the results and/or discussion sections. In the introduction the authors should report the aims of the work and not the results obtained, i.e. “The expression level of several stress-responsive genes in tea microplants was determined to unravel the correlations between physiological and genetic responses under in vitro mannitol treatment”.
- Thank you for the suggestion. We corrected this point.
Lines 122-124: This sentence is speculative because the authors did not measure the water potential.
- corrected
Line 161: Please, replace “negatively” with “negative”
- done
Line 225: Please, replace “activate” with “activates”
- done
Line 232: Please, separate “raffinose” and “synthesis”
- done

Reviewer 2 Report
Dear authors,
I read your manuscript and I consider that it is very interesting, however, I think that you need to improve several aspects related with the description of background, methods and results, discussion and obtained conclusions. Next, I indicate my major and minor concerns about you manuscript.
Major concerns:
- I consider that the title could be improved. For instance, you results contain biochemical and molecular information, but it is not included in the title.
- I think that the abstract lacks of information about the ‘biological problem’. What is the biological problem triggering this study? Why is it important to know physiological, biochemical of molecular responses to drought stress in tea microplants? You could include 1-2 brief sentences at the beginning of the abstract.
- Units of parameters. Please, review ‘Mmol’ (I suppose you want to use ‘mM), units of light intensity (lux should be substituted by µmol m-2 s-1, among others.
- You include some sentences about results, but you do not show these results. For instance, lines 105 and 126-127. You must show all data.
- You must order your figure panels in the same order to the description of results and vice versa.
- Drought-related genes. You include the relative expression of 31 genes, however, there are some key TFs acting as ‘master regulators’ that are not included in your results (https://www.frontiersin.org/articles/10.3389/fpls.2014.00170/full). Some of these crucial TFs are included like CBF1 or some NAC TFs, but I could not found others. I think that you should justify very well why you use these TFs or include expression of this key TFs.
- Figure legends. Figure captions should contain more information.
- From my point of view, discussion section should be improved. For instance, I think that a brief introduction is necessary. Besides, results should be discussed more deeply respect to previous findings of other authors. I think that this section can be reviewed in depth when these changes are made.Please, avoid the term ‘suppose’.
- Materials and methods. More information about methods is necessary to a better understanding. For instance, (1) I think that the formula RWC is not correct compared with the reference {51}. (2) more information about pigment extraction, HPLC analyses and PCR conditions should be provided.
- I think that some conclusions are incomplete and others could be improved according your results.
- What is the meaning of ‘positively or negatively correlated’?
Minor concerns:
- Lines 40-42, 95-7. This sentence is not well understood.
- Line 58. Add ‘molecular’?
- Line 59-61. I think that this sentence is part of the beginning of paragraph lines 72-82.
- You use ‘stress tolerance’, ‘stress-related genes’,… but you should indicate that it is related to ‘drought’… For instance, line 63, 134,… Please, review through the manuscript.
- Line 68. Add ‘and’ between ARR and SPL.
- Line 90. ‘resulted significant elevation of electrolyte leakage’ but, what is this data? Line 91. ‘EC’ or ‘REC’?
- Line 92. What is the meaning of ‘nonlinear tendency’? I think that this term is not correct.
- You should indicate if results of chloropylls, amino acids content, and so on, corresponds to FW or DW. Please, also modify in graphs.
- Figure 1. The top of the picture is not informative.
- Figure 2. Include units of mannitol concentration e include FW or DW. The same for Figure 3.
- Line 105, 126-127. What is this data?
- Lines 107-109. I suggest use ‘fold change’ to describe these results.
- Figure 5. What is the meaning of ‘orange’ color?
- Line 283. Include the City and Country.
- Concentration unit: ‘mg/l’ or ‘mg/L’?
- Line 287-288. How old are the plants?
- Line 289. ‘Control media (1/2 MS…’ by ‘Control medio composed by ½ MS…’ or something like that.
- Line 293. Please, remove ‘+’.
- Line 307. I think that this formula is correct but it does not match with the reference 51. Please remove ‘%’ and add it to RWC as: RWC (%).
- Line 320. ‘Toral RNA was extracted’ from leaves? And how much quantity of tissue was used?
- Lines 327. Please, indicate that supplementary table 1 contain the primer list of genes, and remove primer sequences of actin.
- Statistical analysis. Did you use technical replicates’? Please, include it.
Author Response
Dear Reviewer, Thank you for the careful examination of our results. We are really appreciating this. We agree with all suggestions and we are really grateful for your work with our article. We have tried to do your best to follow your suggestions. We increase the reference list by more than 25 new references and compared our results with them. We have tried improved introduction and discussion parts as it was suggested. However, if the reviewers are not fully satisfied with our responses, we are ready to complete or revise the manuscript further.
I consider that the title could be improved. For instance, you results contain biochemical and molecular information, but it is not included in the title.
- Thank you for the suggestions. We have tried to improve the title
I think that the abstract lacks of information about the ‘biological problem’. What is the biological problem triggering this study? Why is it important to know physiological, biochemical of molecular responses to drought stress in tea microplants? You could include 1-2 brief sentences at the beginning of the abstract.
- We have added the sentence in the beginning of the Abstract.
Units of parameters. Please, review ‘Mmol’ (I suppose you want to use ‘mM), units of light intensity (lux should be substituted by µmol m-2 s-1, among others.
- Done
You include some sentences about results, but you do not show these results. For instance, lines 105 and 126-127. You must show all data.
- I have tried to revise it slightly. If you insist we can add more graphs with nonsignificant changes in 8 amino acids. However, we suppose that is will overload the MS with no-interesting illustrations
You must order your figure panels in the same order to the description of results and vice versa.
- Done
Drought-related genes. You include the relative expression of 31 genes, however, there are some key TFs acting as ‘master regulators’ that are not included in your results (https://www.frontiersin.org/articles/10.3389/fpls.2014.00170/full). Some of these crucial TFs are included like CBF1 or some NAC TFs, but I could not found others. I think that you should justify very well why you use these TFs or include expression of this key TFs.
- Dear editor, thank you for your suggestion and for the article. I’ve added it to the reference also.
We selected just 31 genes for our study. I understand, that ideally, full transcriptome analysis should be performed to cover all the genes involved in response to the certain osmotic stress factor. For this current study we selected the genes which were earlier supposed as the possible genetic markers of cold or drought tolerance in tea plant [14-20] and which were proved to be involved in osmotic stress responses in Caucasian tea genotypes , which we have been studying . Full transcriptome analysis is our next goal.
Figure legends. Figure captions should contain more information.
- Added
From my point of view, discussion section should be improved. For instance, I think that a brief introduction is necessary. Besides, results should be discussed more deeply respect to previous findings of other authors. I think that this section can be reviewed in depth when these changes are made. Please, avoid the term ‘suppose’.
- We have tried to follow your suggestion and improve this part.
Materials and methods. More information about methods is necessary to a better understanding. For instance, (1) I think that the formula RWC is not correct compared with the reference {51}. (2) more information about pigment extraction, HPLC analyses and PCR conditions should be provided.
- We agree with this comment. We changed the reference # 51. We have completed the MM with more details.
I think that some conclusions are incomplete and others could be improved according your results.
What is the meaning of ‘positively or negatively correlated’?
- Positively correlated genes are the genes which have similar expression character through the treatments. Negative correlation means the opposite expression character through the treatments. Evaluation of correlations between the genes and downstream biochemical processes can help identify gene modules whose expression profiles are very similar. It can help better understanding the mechanism of the response and adaption of crops to each stress factor. The modules of densely inter-connected genes can be analyzed by searching patterns in connection strength [Sharma et al. 2018]. High level of correlation indicates high strength connection of the genes which can be co-expressed or can be linked to the joint regulons. This information can be provided through the analysis of the expression profiles of the each gene through the different treatments.
Lines 40-42, 95-7. This sentence is not well understood.
- We have tried to improve it
Line 58. Add ‘molecular’?
- done
Line 59-61. I think that this sentence is part of the beginning of paragraph lines 72-82.
- we are agree
You use ‘stress tolerance’, ‘stress-related genes’,… but you should indicate that it is related to ‘drought’… For instance, line 63, 134,… Please, review through the manuscript.
- We corrected these points through the whole body of the MS
Line 68. Add ‘and’ between ARR and SPL.
- done
Line 90. ‘resulted significant elevation of electrolyte leakage’ but, what is this data? Line 91. ‘EC’ or ‘REC’?
- we clarified this point
Line 92. What is the meaning of ‘nonlinear tendency’? I think that this term is not correct.
- we corrected this statement
You should indicate if results of chloropylls, amino acids content, and so on, corresponds to FW or DW. Please, also modify in graphs.
- Done
Figure 1. The top of the picture is not informative.
- Removed
Figure 2. Include units of mannitol concentration e include FW or DW. The same for Figure 3.
- Done
Line 105, 126-127. What is this data?
- Clarified
Lines 107-109. I suggest use ‘fold change’ to describe these results.
- Revised according to your advice
Figure 5. What is the meaning of ‘orange’ color?
- Orange and light blue color = nonsignificant positive and negative correlations, respectively.
Line 283. Include the City and Country.
- done
Concentration unit: ‘mg/l’ or ‘mg/L’?
- Corrected
Line 287-288. How old are the plants?
- The plants were propagated for 2 years before the experiment. The experimental plants were grown for 2 month on the experimental media. I have add details in MM.
Line 289. ‘Control media (1/2 MS…’ by ‘Control medio composed by ½ MS…’ or something like that.
- Revised accordingly.
Line 293. Please, remove ‘+’.
- Done
Line 307. I think that this formula is correct but it does not match with the reference 51. Please remove ‘%’ and add it to RWC as: RWC (%).
- Done
Line 320. ‘Toral RNA was extracted’ from leaves? And how much quantity of tissue was used?
- Clarified in methods
Lines 327. Please, indicate that supplementary table 1 contain the primer list of genes, and remove primer sequences of actin.
- Done
Statistical analysis. Did you use technical replicates’? Please, include it.
- Done
Reviewer 3 Report
The comments are presented below:
Introduction - Lack of information connecting osmotic stress and drought. It seems to me that the authors equated these two terms and use them interchangeably, what is obviously incorrect. Please explain in this section why you used mannitol, and not the other osmotically active substances.
In methods - please include more details on physiological analyzes - there is a lack bunch of information making hard to evaluate correctness of the methodology i.e. REC analyses – did you used whole leaves, or leaf discs?, how many leaves/discs were used? was vacuum applied? How many biological and technical repetitions were taken? What was the temperature during measurements etc. Please go through all your methods and change it accordingly. The same in gene expression analysis, if you claim that you modified CTAB method you should explain how it was changed.
Results and Discussion
Line 87 it seems to me (looking at the picture) you observed necrosis, as it is located only at the tip of the leaf.
It seems from the methods, that authors measured chlorophyll a and b, why results are presented for sum of chlorophylls?
Figs 4 and 5 should be of better quality, as it is hard to read them.
Line 177-179 is a speculation, as ROS were not measured in this study
Line 189 – this info should be places in the results section, not discussion
Lines 191-206 The results of biochemical study is not discussed deeply. There are many papers dealing with the accumulation or degradation of certain chemical during various kind of stresses and Authors should look deeper. The conclusions (i.e lines 197, 206) should be omitted.
Line 215 – I suppose authors meant chaperones?
Line 236 cold stress – Please insert citation
Line 253 speculation as glucose was not tested.
Author Response
Dear Reviewer,
thank you for the careful examination of our results. We are really appreciating this. We agree with all suggestions and we are really grateful for your work with our article. We have tried to do your best to follow your suggestions. We increase the reference list by more than 25 new references and compared our results with them. We have tried improved introduction and discussion parts. However, if you are not fully satisfied with our responses, we are ready to complete or revise the manuscript further.
Introduction - Lack of information connecting osmotic stress and drought. It seems to me that the authors equated these two terms and use them interchangeably, what is obviously incorrect. Please explain in this section why you used mannitol, and not the other osmotically active substances.
- We have tried to follow your suggestions and improve this part.
In methods - please include more details on physiological analyzes - there is a lack bunch of information making hard to evaluate correctness of the methodology i.e. REC analyses – did you used whole leaves, or leaf discs?, how many leaves/discs were used? was vacuum applied? How many biological and technical repetitions were taken? What was the temperature during measurements etc. Please go through all your methods and change it accordingly. The same in gene expression analysis, if you claim that you modified CTAB method you should explain how it was changed.
- Dear editor we agree with your suggestions. I have completed MM part with more details. Hope now it is enough information. However, if something is still missed we are ready to complete.
Line 87 it seems to me (looking at the picture) you observed necrosis, as it is located only at the tip of the leaf.
- Yes, some plants but not all have necrosis on the leaf tips in the presence of Mannitol 300 mM. No necrosis was observed in control and Mannitol 200
It seems from the methods, that authors measured chlorophyll a and b, why results are presented for sum of chlorophylls?
- Added
Figs 4 and 5 should be of better quality, as it is hard to read them.
- We have tried to improve the images
Line 177-179 is a speculation, as ROS were not measured in this study
- removed
Line 189 – this info should be places in the results section, not discussion
- Rephrased
Lines 191-206 The results of biochemical study is not discussed deeply. There are many papers dealing with the accumulation or degradation of certain chemical during various kind of stresses and Authors should look deeper.
- We have tried to improve this part according to your suggestions
The conclusions (i.e lines 197, 206) should be omitted.
- done
Line 215 – I suppose authors meant chaperones?
- revised
Line 236 cold stress – Please insert citation
- done
Line 253 speculation as glucose was not tested.
- The sentence is removed

Round 2
Reviewer 1 Report
The authors have improved the work throughout the manuscript by taking into account most of the reviewers' suggestions. I have only a few other minor comments and I suggest the authors to look carefully for grammatical errors.
Line 112: Please, delete “We”
Lines 113-114:I suggest to change the sentence as: “These genes have also been shown to be involved in cold responses in Caucasian tea genotypes”
Line 135: Please, replace “the was” with “there was”
Figure 2: Please, indicate the treatments in the same way in the different panels and throughout the manuscript (i.e. with -panel A and B- or without -panel C and D- space between “Mannitol” and the concentration level).
Line 173: Please, replace (Fig 2C, Fig3C, D) with (Fig. 2C, Fig. 3C,D).
Line 178: caption of Fig. 3, please replace “tea fresh leaves” with “fresh tea leaves”.
Line 184: Please replace “thirty one” with “thirty-one”.
Line 243: Please, replace “the higher” with “the highest”.
Line 252: The assessment of biochemical compounds in fresh leaves has disadvantage, not only because it may be diluted by a higher plant biomass of the leaves, but also because of the differences in water amount (and hence the dilution effect) within tissue, especially when comparing fully hydrated and partially dried leaf tissues as in this case. I agree with the authors that for many biochemical compounds it is necessary to work on the fresh material to avoid their oxidation and degradation; however, in these cases it is possible to take a sub-sample on which to make the percentage of dry matter in order to compare the data expressed on both fresh and dry weight.
Line 279: Please, replace “arabidopsis” with “Arabidopsis”
Line 281: Please, replace “propose” with “proposed”
Line 302: Please, add “who” before “showed”
Line 439: Please, correct “spectophotometrically” with “spectrophotometrically”
Author Response
Dear reviewer,
thank you very much for your useful comments and suggestions.
Line 112: Please, delete “We”
- deleted
Lines 113-114:I suggest to change the sentence as: “These genes have also been shown to be involved in cold responses in Caucasian tea genotypes”
- changed
Line 135: Please, replace “the was” with “there was”
- replaced
Figure 2: Please, indicate the treatments in the same way in the different panels and throughout the manuscript (i.e. with -panel A and B- or without -panel C and D- space between “Mannitol” and the concentration level).
- I’m not sure that I understood this comment right, but I have tried to verify the letters and the spaces in the figures and through the MS
Line 173: Please, replace (Fig 2C, Fig3C, D) with (Fig. 2C, Fig. 3C,D).
- replaced
Line 178: caption of Fig. 3, please replace “tea fresh leaves” with “fresh tea leaves”.
- Revised accordingly
Line 184: Please replace “thirty one” with “thirty-one”.
- Done
Line 243: Please, replace “the higher” with “the highest”.
- Replaced
Line 252: The assessment of biochemical compounds in fresh leaves has disadvantage, not only because it may be diluted by a higher plant biomass of the leaves, but also because of the differences in water amount (and hence the dilution effect) within tissue, especially when comparing fully hydrated and partially dried leaf tissues as in this case. I agree with the authors that for many biochemical compounds it is necessary to work on the fresh material to avoid their oxidation and degradation; however, in these cases it is possible to take a sub-sample on which to make the percentage of dry matter in order to compare the data expressed on both fresh and dry weight.
- This clarification is inserted into the text of discussion.I agree with this suggestion. In our future work we will follow this advice.
Line 279: Please, replace “arabidopsis” with “Arabidopsis”
- Replaced
Line 281: Please, replace “propose” with “proposed”
- Replaced
Line 302: Please, add “who” before “showed”
- Added
Line 439: Please, correct “spectophotometrically” with “spectrophotometrically”
- Corrected

Reviewer 2 Report
Dear authors,
First, I want to thank you for your efforts to improve this manuscript. I think that the manuscript has improved subtantially compared to the previous version.
Second, I have two concerns about the new version:
1) Accordingly to the previous revision, I ask them to show non-significant results as well. I did not give you any reasons about that, but I give you several reasons to understanding this concern. From my point of view, the 'negative' results, also are necessary to understanding the research. In addition, when other authors read your paper, they may compare their results with yours. Finally, I think that the showing of full results is a good practice for transparency of science, avoinding that readers have doubts about your results. You could remember that: 'Negative results are good results too'. Anyway, this is my point of view.But, I think that is very important to visualize these results, and you could include it in supplementary information.
2) Please, indicate 'mM' unitos for mannitol concentration in graphs (200 and 300 mM).
3) Please, include it the amplicon size for RT-qPCR primers in supplementary table.
Author Response
Dear Reviewer,
Thank you very much for your useful comments and suggestions.
Accordingly to the previous revision, I ask them to show non-significant results as well. I did not give you any reasons about that, but I give you several reasons to understanding this concern. From my point of view, the 'negative' results, also are necessary to understanding the research. In addition, when other authors read your paper, they may compare their results with yours. Finally, I think that the showing of full results is a good practice for transparency of science, avoinding that readers have doubts about your results. You could remember that: 'Negative results are good results too'. Anyway, this is my point of view. But, I think that is very important to visualize these results, and you could include it in supplementary information.
- We agree with this advice. We have attached the Supplementary file 1 with amino acids data.
2) Please, indicate 'mM' unitos for mannitol concentration in graphs (200 and 300 mM).
- indicated
3) Please, include it the amplicon size for RT-qPCR primers in supplementary table.
- included

Reviewer 3 Report
The re-submitted version is significantly improved, however some parts needs to be corrected
Line 148-152 Please re-phrase to make it sound more professional
Line 320 – remove the sentence on heavy stress and cell membranes, as is not related to what was studied
Conclusions need to be re-writed. Conclusion (i) is it true? does this metabolites do not accumulate in response to osmotic stress induced by excessive salinity also? – please do not generalize, just state on what is proven by your work
Conclusions (ii -iv) – repetition of results not a conclusion. Try to find a connection between the genes and the stress applied, what these genes are responsible for- how this can be related to osmotic stress
Author Response
Dear Reviewer,
Thank you very much for your useful comments and suggestions. We have tried to follow your advices.
Line 148-152 Please re-phrase to make it sound more professional
- I have tried to do this.
Line 320 – remove the sentence on heavy stress and cell membranes, as is not related to what was studied
- Removed
Conclusions need to be re-writed. Conclusion (i) is it true? does this metabolites do not accumulate in response to osmotic stress induced by excessive salinity also? – please do not generalize, just state on what is proven by your work
- Yes , we agree with this comment. I have tried to revise this sentence
Conclusions (ii -iv) – repetition of results not a conclusion. Try to find a connection between the genes and the stress applied, what these genes are responsible for- how this can be related to osmotic stress
- We have tried to revise the conclusions according to your suggestion.
